# Enzymatic Biotransformation of *13-desmethyl Spirolide C* by Two Infaunal Mollusk Species: The Limpet *Patella vulgata* and the Cockle *Cerastoderma edule*

**DOI:** 10.3390/toxins14120848

**Published:** 2022-12-02

**Authors:** Araceli E. Rossignoli, Juan Pablo Lamas, Carmen Mariño, Helena Martín, Juan Blanco

**Affiliations:** 1Centro de Investigacións Mariñas (CIMA), Pedras de Corón s/n, 36620 Vilanova de Arousa, Spain; 2Intecmar (Instituto Tecnolóxico para o Control do Medio Mariño de Galicia), Peirao de Vilaxoán s/n, Vilagarcía de Arousa, 36611 Pontevedra, Spain

**Keywords:** biotransformation, isomerization, metabolites, mollusks, spirolides

## Abstract

The presence of a 13-desmethyl Spirolide C isomer (Iso-13-desm SPX C) is very common in some infaunal mollusks in Galicia contaminated with this toxin. Its possible origin by biological transformation was investigated by incubating homogenates of the soft tissues of limpets and cockles spiked with 13-desmethyl Spirolide C (13-desm SPX C). The involvement of an enzymatic process was also tested using a raw and boiled cockle matrix. The enzymatic biotransformation of the parent compound into its isomer was observed in the two species studied, but with different velocities. The structural similarity between 13-desm SPX C and its isomer suggests that epimerization is the most likely chemical process involved. Detoxification of marine toxins in mollusks usually implies the enzymatic biotransformation of original compounds, such as hydroxylation, demethylation, or esterification; however, this is the first time that this kind of transformation between spirolides in mollusks has been demonstrated.

## 1. Introduction

Spirolides (SPXs) were first discovered in the 1990s in the Atlantic coast of Nova Scotia, Canada, when unusual toxicities were detected by mouse bioassay in mussel and scallop extracts [1]. They are currently detected worldwide [2,3,4,5,6,7,8]. To date, only the dinoflagellate *Alexandrium ostenfeldii* (sometimes cited as *A. peruvianum*) [5,9] has been identified as a producer of SPXs. Spirolides belong to the cyclic imine group (CIs) of neurotoxins, including gymnodimines, pinnatoxins, pteriatoxins, portimines, spiro-prorocentrimines, and prorocentrolides. SPXs constituted the largest group of CIs. Currently, numerous SPXs-analogues are known in European, Oceanian, Asian, and American waters [10,11,12,13,14,15], with 13-desmethyl SPX C (13-desm SPX C) being the most commonly found in shellfish (Figure 1). SPXs are responsible for a fast-acting toxicity after intraperitoneal injection (i.p.) in mice, [16] produce neuronal damage in the hippocampus and brain stem of mice and rats after i.p. injection with 13-desm SPX C, and increase mRNA levels of muscarinic and nicotinic acetylcholine receptors. Some studies have indicated that the toxicity of SPXs after i.p. injection is higher than that after oral administration [17,18,19], which is consistent with the absence of harmful effects in humans consuming shellfish contaminated with these toxins (except non-specific symptoms, such as gastric distress and tachycardia) [20]. For these reasons, nowadays, there is no regulatory limit for spirolides in shellfish in Europe or other regions of the world. The European Food Safety Authority (EFSA), nevertheless, has requested more data to properly assess the risk that CIs in general and SPXs in particular pose to shellfish consumers, and has only suggested a tentative maximum limit alert of 400 µg of total SPXs kg^−1^ shellfish meat [21].

CIs can be transformed into bivalves; however, the information available regarding their metabolism is very limited. Some of these compounds can undergo reduction (gymnodimines), ring-opening (SPX A and B), or spirolide esterification in mollusks [22,23,24,25,26]. The epimerization of one spirolide (20-methyl spirolide G) by mussels has been suggested by Aasen et al. [6] in view of the differences between the toxin profiles of the bivalve and the phytoplankton from the same area.

The toxin 13-desm SPX C was frequently detected in the analyses of lipophilic toxins routinely performed in the Official Galician Monitoring System carried out by Instituto Tecnolóxico para o Control do Medio Mariño (INTECMAR). In addition, a second peak with the same *m*/*z* and a retention time close to 13-desm SPX C was detected. The fragmentation pattern (MS2 and MS3) of this second compound was practically identical to that of 13-desm SPX C, indicating that the second peak was an isomer. According to analyses carried out by INTECMAR, this isomer is very common in most infaunal mollusk species from the Galician coasts, such as cockles, abalone, and some clams, but not in mussels (wild or cultured in rafts). The source of this isomer is unknown, but biotransformation from 13-desm SPX C seems to be most likely. In this work, we investigated whether 13-desm SPX C is the origin of its isomer in two mollusk species and whether it is generated by means of an enzymatic transformation.

## 2. Results

### 2.1. Transformation Kinetics in the Limpet Patella vulgata

The results obtained in the limpets clearly showed that the concentration of 13-desm SPX C decreased with the incubation time, whereas the peak corresponding to its isomer, Iso-13-desm SPX C, increased (Figure 2). The increase in the isomer was approximately linear, but the decrease in 13-desm SPX C appeared to be asymptotic.

The recorded total change of 13-desm SPX C with the incubation time was much higher than that of its isomer.

### 2.2. Transformation Kinetics in the Cockle Cerastoderma edule

In the case of cockles (*C. edule*), there was an obvious increase in the concentration corresponding to the peak of Iso-13-desm SPX C during incubation. However, contrary to what happened in limpets, the decrease in 13-desm SPX C in cockles was also linear. In this species, the rate of production of the isomer was approximately equivalent to the decrease in 13-desm SPX C, as shown by the slopes of both regressions, which were practically the same but with opposite signs (Figure 3).

### 2.3. Effect of Boiling

Boiling substantially suppressed the production of Iso-13-desm SPX C during incubation. After incubation, the concentration of this compound was approximately 12 times higher in the raw samples but only 2.5 times higher in the boiled samples (Figure 4). Boiling had no significant effect on the change in the concentration of 13-desm SPXC after 22 h of incubation. A small decrease in concentration was detected during incubation with the boiled homogenate (the median concentration was approximately 90% of that corresponding to incubation with the raw extract).

## 3. Discussion

Iso-13-desm SPX C originates from the transformation of 13-desm SPX C in both the gastropod *Patella vulgata* and the bivalve *Cerastoderma edule*, but probably also in other mollusks. To date, as far as we know, this is the first work that describes the existence of transformations between spirolides in mollusks, even when at least epimerization has been suggested by Aasen et al. [6]. Metabolic transformations in bivalves after the filtration/absorption of toxic algae are frequent. These are the main reasons why the toxic profiles detected in dinoflagellates and shellfish are usually different [27]. Enzymatic mechanisms associated with the detoxification of toxins in bivalves are very common in lipophilic toxins and have been previously documented. Esterification with fatty acids, as in the case of okadaic acid (OA) [28,29,30,31] and spirolides [25,26], or opening macro-ring structures, such as in pectenotoxin 2 (PTX-2) [32,33,34], and some oxidations are frequent, but not isomerizations.

The fragmentation spectrum obtained in the Iso-13-desm SPX C using double-(MS2) and triple-stage mass spectrometry (MS3) showed fragmentation products that were very similar to the original toxin (in prep.). This seems to indicate that the isomer obtained by enzymatic transformation in mollusks may correspond to an epimer of 13-desm SPX C. Epimers of PSP [35], DSP toxins, pectenotoxins [36], domoic acid [37], and some cyclic imines, such as pinnatoxins B/C [38,39] and gymnodimine B/C [40], have been documented in bivalves and have not, in some cases, been linked to transformations in invertebrates. To the best of our knowledge, this is the first time that a transformation that led to the production of a possible epimer of a spirolide has been reported.

The transformation of 13-desm SPX C into Iso-13-desm SPX C was observed in the two species studied, but with slight differences between them. In the limpets, the decrease in the concentration of 13-desm SPX C was notably higher than the increase in its corresponding isomer. The possibilities of the existence of a matrix effect or loss of sensitivity of the isomer were ruled out, and no other new spirolide was detected. Therefore, it seems obvious that an important degradation of the original spirolide 13-desm SPX C in this gastropod took place. For cockles, the rate of isomerization indicates that this species converts 13-desm SPX C into its isomer more quickly than limpets; however, in this case, there was no degradation of the parent toxin. We do not have any explanation for these differences, although the fact that they are two classes of mollusks (bivalve and gastropod) with different habitats, feeding patterns and physiologies may be in some way related.

The isomerization demonstrated in this work appears to be an enzymatic process because boiling the shellfish (which denatures proteins) and the absence of protease inhibitors (which allows their degradation) substantially decrease the production of Iso-13-desm SPX C during incubation in cockles. Even when the microbial origin of the responsible enzymes cannot be discarded, three facts render it unlikely. The first one is that the organisms used in this experiment were maintained for 7 days in filtered seawater, which substantially reduces the organic contents of the gut and, hence, the microbial load. The second is that large differences in transformation among different bivalve species sharing the same habitat have been found (unpublished observations). The third is that spirolides are accumulated intracellularly, which would require intracellular bacteria to carry out the transformations recorded in natural bivalve populations.

Information on the possible sites of isomerization of cyclic imines is scarce. Pinnatoxins B and C have different steric configurations of carboxylic and amino groups located in the position occupied by the butenolide group of the spirolides. The steric configuration of this group seems to be constant for the main CIs [41], which suggests a biosynthetically conserved relative configuration at C4 among these toxins. However, Kong et al. [42] were able to synthesize diastereomer C4-epi-gymnodimine from natural gymnodimine A, which suggests that compounds with this alternative configuration are possible, even when they are not synthetized by phytoplankton. Pinnatoxins B and C also differ in the steric configuration of the butanolide group [43]. Gymnodimine A and B differ in the location of a double bond, which may not substantially affect the fragmentation pattern, in view of the pathways suggested by Sleno et al. [44]. Therefore, one of these two possibilities seem to be the most likely isomerization mechanism for 13-desm SPX C. In contrast, in 2008, Christian et al. [45] incubated extracts from *Alexandrium ostenfeldii* (containing 13,19-didesm SPX C, 13-desm SPX C, and SPX G) with enzymatic cell-free extracts from three bivalve species (mussels, oysters, and scallops) and did not show any evidence of the possible metabolization of the three spirolides, such as ring opening, hydroxylation, demethylation, or other conversions. We do not know if 13-desm SPX C has the same toxic power as its isomer; hence, the isomerization detected in some species may modify the risk that their consumption poses for human health. It is necessary to carry out toxicological studies to evaluate the threat that these compounds may pose to shellfish consumers. Biotransformation products following the absorption of marine toxins are constantly detected in invertebrates and highlight the importance of analyzing metabolites when assessing exposure and for effective monitoring programs.

## 4. Materials and Methods

### 4.1. Chemicals and Reference Materials

Acetonitrile (LC-MS grade) and methanol (HPLC grade quality) were purchased from Scharlab (Sentmenat, Spain) and VWR (Llinars del Vallés, Spain), respectively. Ultrapure water was obtained from a Milli-Q gradient system fed with an Elix Advantage-10 (Millipore Ibérica, Madrid, Spain). Ammonium hydroxide (NH_4_OH, 25%) was obtained from Merck (Barcelona, Spain) and SIGMAFAST Protease Inhibitor Tablets from Sigma-Aldrich (Madrid, Spain). Quality-controlled standard (QCS) for 13-desmethyl SPX C was obtained from CIFGA, S.A. (Lugo, Spain).

### 4.2. Shellfish Samples and Experimental Design

Limpets (*Patella vulgata*) were obtained from Corón, Vilanova de Arousa, Spain (42°34′30″ N 8°49′50″ W), and cockles (*Cerastoderma edule*) from Praia da Virxe, Muros, Spain (42°46′58″ N 9°03′55″ W). The two species were selected because they are the ones in which the highest concentrations of the isomer have been observed and because they belong to two classes of mollusks (bivalve and gastropod) with different habitats (rocky epifaunal and sandy infaunal), feeding patterns and physiologies. In both cases, individuals were maintained in 50 L tanks with running seawater at approximately 18 °C for 6–7 days previous to the experiments to remove any possible spirolide remaining in the gut, reduce the bacterial load, acclimatize them, and recover all their normal biological and physiological activities before the experiment.

Three experiments were performed. Two of them aimed at spirolide transformation in the mollusk species, and the third aimed to check the effect of protein denaturation on transformation. In the transformation experiments, the shells of the mollusks were removed, and approximately 20 g of soft tissue mixed with two protease inhibitor tablets dissolved in 1 mL of water was homogenized with an Ultraturrax T25 (IKA, Staufen, Germany). The obtained homogenate was rapidly distributed, at random, in 27 aliquots (0.5 g) in Eppendorff tubes (5 mL). To reduce the viscosity of the homogenate and favor the distribution of 13-desm SPXC addition, 100 µL of Milli-Q water was added to each aliquot. Three aliquots (Initial) were used to evaluate the base content of 13-desm SPX C and Iso-13-desm SPX C, which were not spiked with any compound. The remaining aliquots (24) were spiked with 19 µL of 13-desm SPX C (concentration 7 µg mL^−1^), vortexed for 30 s, and homogenized for 5 min in an ultrasonic bath filled with a mixture of water and ice. The first set of three aliquots was taken immediately, and each aliquot was mixed with 2 mL of MeOH and frozen at −20 °C until extraction. All other aliquots were maintained under constant agitation at 23 °C for 2, 4, 6, 8, 10, 12 and 22 h. After their corresponding incubation time, one set of three aliquots was removed from the shaker and subjected to the same process as in the initial set. After 4–5 days, all samples were thawed and extracted in an ultrasonic bath filled with a mixture of water and ice. Finally, the obtained extracts were clarified by centrifugation at 19,000× *g*, filtered through 0.22 µm PES syringe filters, and analyzed by liquid chromatography coupled to mass spectrometry (LC-MS/MS).

Additionally, a short experiment was performed to check whether the transformation had an enzymatic origin. In this case, 5 g of cockle tissue (*Cerastoderma edule*), without the addition of a protease inhibitor cocktail, was boiled for 5 min and subsequently cooled in an ultrasonic bath filled with a mixture of water and ice for 5 min. Six 0.5-g aliquots of this boiled homogenate were spiked with 5 µL of 13-desm SPX C (concentration 7 µg·mL^−1^), vortexed for 30 s, and homogenized in an ultrasonic bath filled with a mixture of water and ice. Three aliquots were processed at time 0, and the other three were incubated for 22 h and extracted with 100% methanol, as explained above.

### 4.3. LC-MS/MS Procedure

Analyses were carried out on an Exion LC AD™System (SCIEX, Framingham, MA, USA) coupled to a Qtrap 6500+ mass spectrometer (SCIEX) through an IonDrive Turbo V interface in electrospray mode. The toxins were separated according to a previously described method [46]. The mass spectrometer parameters were set as follows: ion source Gas 1, 75 (arbitrary units); ion source Gas 2, 75 (arbitrary units); ion spray voltage positive, 5000; heater gas temperature, 600 (°C); curtain gas, 30; collision Gas, medium. The transitions 692.5 > 164.3 and 692.5 > 444.3, with collision energies of 60 and 55 v, were used as quantifier and qualifier, respectively. 13-desm SPX C and Iso -13-desm SPX C in the extracts were quantified by the external standard method using dilutions of a solution of 13-desm SPX C in methanol, assuming an equal response of the two toxins on a molar basis [2,29,47].

### 4.4. Statistical Analysis

Statistical analyses were performed using the R base and the R Stats packages [48]. Descriptive statistics and associated graphics were generated using ggplot2 [49].

## Figures and Tables

**Figure 1 toxins-14-00848-f001:**
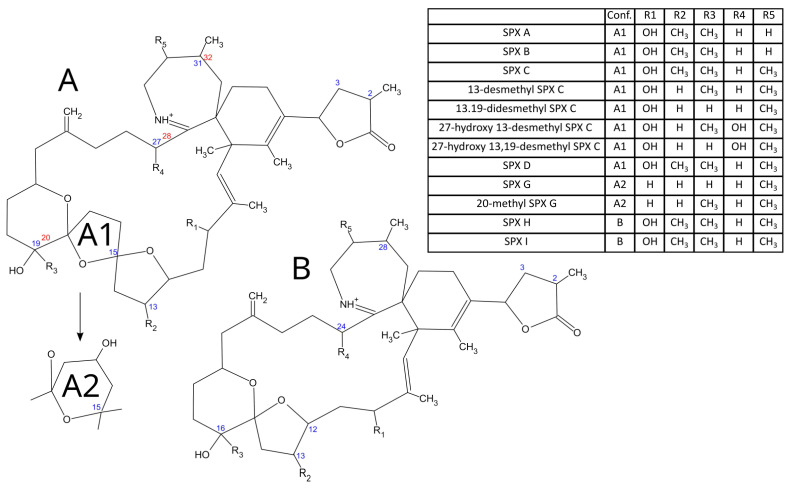
(**A**,**A1**,**A2**,**B**) correspond with the two configuration structures that can present the different types of spirolides.

**Figure 2 toxins-14-00848-f002:**
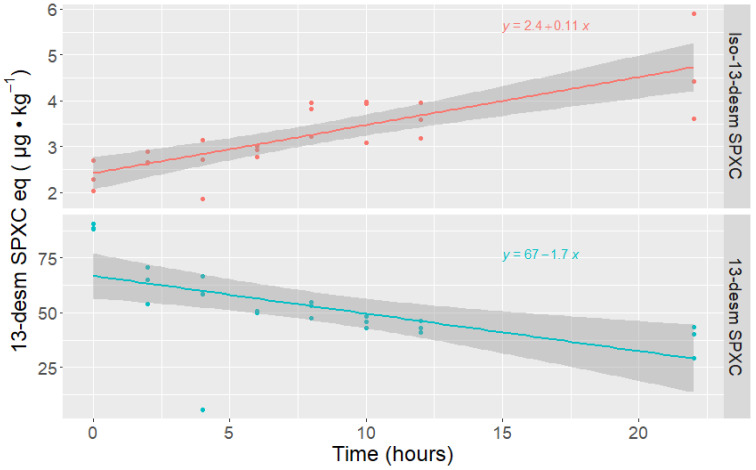
Concentrations of 13-desm SPX C and Iso-13-desm SPX C along the incubation of 13-desm SPX C in limpet (*Patella vulgata*) homogenates.

**Figure 3 toxins-14-00848-f003:**
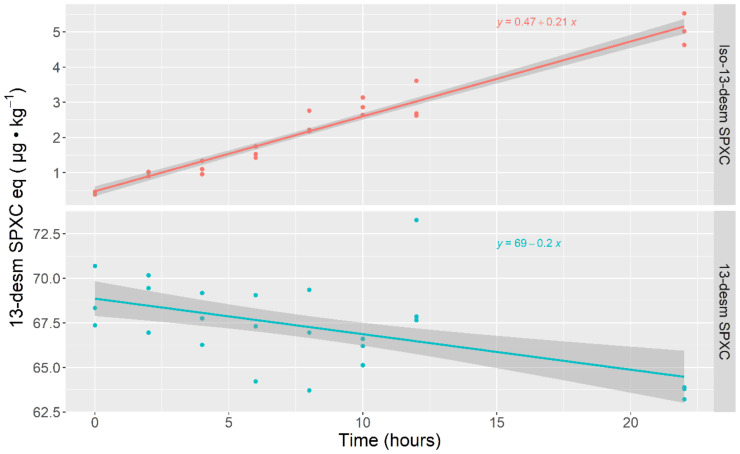
Concentrations of 13-desm SPX C and Iso-13-desm SPX C along the incubation of 13-desm SPX C in cockle (*Cerastoderma edule*) homogenates.

**Figure 4 toxins-14-00848-f004:**
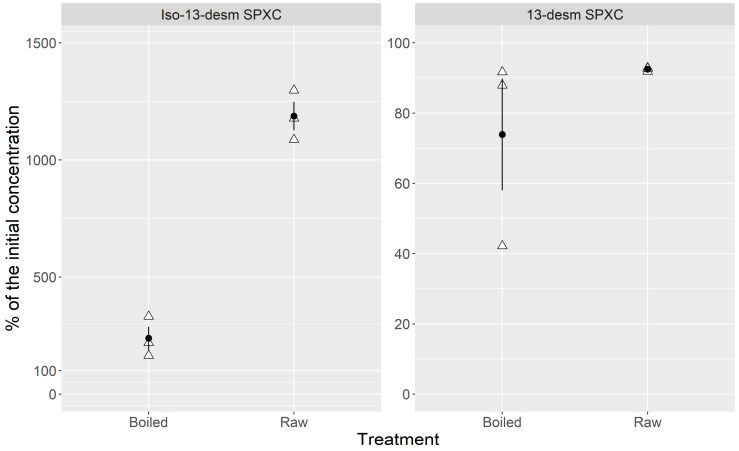
Means (circles), their standard errors (vertical lines) and individual observations (triangles) of the percentages of the initial concentration of 13-desm SPX C and Iso-13-desm SPX C after a 22 h incubation of 13-desm SPX C in raw and boiled cockle homogenates.

## Data Availability

Data are available on request in the Centro de Investigacións Mariñas.

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
