# Peer review of "Enzymatic Biotransformation of 13-desmethyl Spirolide C by Two Infaunal Mollusk Species: The Limpet Patella vulgata and the Cockle Cerastoderma edule"

_toxins, 2022, doi:10.3390/toxins14120848_

Round 1

Reviewer 1 Report

Manuscript: 2035723-peer-review-v1

Article: Enzymatic biotransformation of 13-desmethyl spirolide C by two infaunal mollusk species

A.               This is a very interesting work.

B.               Some comments:

1.      What are the reasons why the two species (limpet Patella vulgata and the cockle Cerastoderma edule) were selected.

2.      The individuals were maintained in 50 L tanks with running seawater at approximately 18 °C for 6-7 days previously the experiments. This step was performed to run some kind of depuration? Or another objective?

3.      Did the material recovered from molluscs only consist of the soft tissue or did it also include the intervalvular fluid?

4.      Have the different steps followed in the experimental design been previously tested and validated? Were they based on other articles or previous trials?

5.      Do the authors have any explanation for the different behaviour of the two species in what regards the formation of the isomer Iso 13-desm SPX C?

Author Response

  1. What are the reasons why the two species (limpet Patella vulgataand the cockle Cerastoderma edule) were selected?

The main reasons were that they are the two species in which the highest concentrations of the isomer have been observed and that they belong to two classes of molluscs (bivalve and gastropod) with different habitats (rocky epifaunal and sandy infaunal), feeding patterns and physiologies

  1. The individuals were maintained in 50 L tanks with running seawater at approximately 18 °C for 6-7 days previously the experiments. This step was performed to run some kind of depuration? Or another objective?

The purpose was to remove any possible spirolide remain in the gut, acclimatize them and to recover all their normal biological and physiological activities before the experiment.

  1. Did the material recovered from molluscs only consist of the soft tissue or did it also include the intervalvular fluid?

We only used soft tissue of the molluscs, without intervalvar fluid. It unlike that intervalvar fluid can contribute significantly to spirolide concentration because of the lipophilic nature of these toxins.

  1. Have the different steps followed in the experimental design been previously tested and validated? Were they based on other articles or previous trials?

A similar incubation designs have been extensively used for studying toxins, as for example:

Shimizu, Y., & Yoshioka, M. (1981). Transformation of paralytic shellfish toxins as demonstrated in scallop homogenates. Science, 212, 547–549.; Sullivan, J. J., Iwaoka, W. T., & Liston, J. (1983). Enzimatic Transformation of PSP toxins in the littleneck clam (Protothaca staminea). Biochemical and Biophysical Research Communications, 114, 465–472.; Fast, M. D., Cembella, A. D., & Ross, N. W. (2006). In Vitro Transformation of Paralytic Shellfish Toxins in the Clams Mya Arenaria and Protothaca Staminea. Harmful Algae, 5(1), 79–90; Jaime, E., Gerdts, G., & Luckas, B. (2007). In Vitro Transformation of PSP Toxins by Different Shellfish Tissues. Harmful Algae, 6(3), 308–316; Oyaneder-Terrazas, J., Polanco, C., Figueroa, D., Barriga, A., & Garcia, C. (2020). In vitro biotransformation of OA-group and PTX-group toxins in visceral and non-visceral tissues of Mytilus chilensis and Ameghinomya antiqua. Food Additives and Contaminants Part a-Chemistry Analysis Control Exposure & Risk Assessment, 37(7); Blanco, J., Marino, C., Martin, H., Alvarez, G., & Rossignoli, A. E. (2021). Characterization of the Domoic Acid Uptake Mechanism of the Mussel (Mytilus galloprovincialis) Digestive Gland. Toxins, 13(7).

  1. Do the authors have any explanation for the different behaviour of the two species in what regards the formation of the isomer Iso 13-desm SPX C?

We don't have any explanation. Although the fact that they are two classes of molluscs (bivalve and gastropod) with different habitats, feeding patterns and physiologies may have some relationship.

Reviewer 2 Report

There is no evidence in this manuscript that Iso-13-desm SPX C is a isomer of 13-desm SPX C and all experiments are based on speculation. The data for the identification of Iso-13-desm SPX C should be provided in the manuscript.

The presence of Iso-13-desm SPX C was inferred in this manuscript, and no data were available to prove that the compound was an isomer of 13-desm SPX C. Therefore, it is impossible to determine whether Iso-SPX really existed or not. Even if Iso-13-desm SPX C was present, only a small amount of Iso-13-desm SPX C was generated, and this conversion was weak. Overall, the study on SPX transformation was lacking in innovation and scientific significance. The journal Toxins, as a high-quality journal, should publish high level papers in the field of toxins.

Author Response

There is no evidence in this manuscript that Iso-13-desm SPX C is an isomer of 13-desm SPX C and all experiments are based on speculation. The data for the identification of Iso-13-desm SPX C should be provided in the manuscript.

It is not the aim of the manuscript to demonstrate that the compound is an isomer of 13-desm SPX C. This demonstration is included in an article with a wider scope (spirolides in the Galician coast) that has been submitted to Toxins. We plan to add a reference to the introduction the manuscript as soon as that article is published.

The presence of Iso-13-desm SPX C was inferred in this manuscript, and no data were available to prove that the compound was an isomer of 13-desm SPX C. Therefore, it is impossible to determine whether Iso-SPX really existed or not.

Here is an extract of the submitted manuscript:

Figure 3. Chromatogram showing the peaks corresponding to 13-desmethyl spirolide C and its isomer for the two fragments routinely used to monitor 13-desmethyl spirolide C.

Figure 4. MS2 fragmentation spectra of 13-desmethyl spirolide C (top panel), its isomer (middle panel) and differences between them (lower panel)

Table 1. Fitting of the obtained MS2 and MS3 spectra of Iso-13-desmethyl spirolide C relative to 13-desmethyl spirolide C as estimated by Sciex Analyst library search.

Spectrum

Collision Energy

Fit

Reverse fit

692.5

60 ± 10

91.1

87.5

692.5 > 444.3

60 ± 10

92.8

92.6

692 > 164.2

60 ± 10

89.7

84.1

Even if Iso-13-desm SPX C was present, only a small amount of Iso-13-desm SPX C was generated, and this conversion was weak.

In our opinion, a transformation rate between 10% and 20% per day (approximately) cannot be considered as “week transformation”, and it is clearly relevant to produce significant concentrations of the isomer in relation to 13-desmSPXC in some mollusks.

Overall, the study on SPX transformation was lacking in innovation and scientific significance. 

As far as we know, there are no published study dealing with biotransformation of spirolides, as we stated in the discussion section (“To date, as far as we know, this is the first work that describes the existence of transformations between spirolides in mollusks, even when, at least epimerization, has been suggested by Aasen et al [6]”).

The journal Toxins, as a high-quality journal, should publish high level papers in the field of toxins.

Our previous reply, in our opinion, fully justify the scientific originality, relevance and the interest to publish our work in a high-quality journal as Toxins.

Round 2

Reviewer 2 Report

1、Microorganisms in shellfish tissues may also promote toxin transformation. How to exclude the role of microorganisms in 13-desm SPX transformation.

2、The experiments had only three replications per treatment, so it is not recommended that Figure 4 be expressed in a box plot, which usually requires at least six data.

3、Why did the 13-desm SPX concentration in the boiled treated samples vary considerably and the minimum concentration was less than 50% of the initial value.

4、Why the concentration of 13-desm SPX in the raw group was lower than that in the boiled group.

Author Response

Reviewer 2

1、Microorganisms in shellfish tissues may also promote toxin transformation. How to exclude the role of microorganisms in 13-desm SPX transformation.

In the manuscript, we have not attributed the enzymatic transformation to enzymes of the mollusk. It could be produced by components of the cockle or by associated microorganisms. Notwithstanding, involvement of microorganism seems to be unlike. The organisms used in these experiments were maintained in filtered seawater for some days to eliminate most of the gut content and, very likely, most microorganisms. The microbial community of different bivalve species living in the same area should be very different to explain the large differences in isomer concentration recorded in the routine monitoring. Additionally, spirolides are accumulated inside the digestive gland cells, which make also difficult to explain the observed difference among living mollusks (These aspects are shown in a manuscript about eight years of spirolide monitoring which has been submitted toToxins).

2、The experiments had only three replications per treatment, so it is not recommended that Figure 4 be expressed in a box plot, which usually requires at least six data.

There is nothing incorrect in using a boxplot because all parameter used to build the plot can be computed from three data. To make the plot clearer, notwithstanding, symbols indicating the precise value of the obtained observations has been added to the Figure 4.

3、Why did the 13-desm SPX concentration in the boiled treated samples vary considerably and the minimum concentration was less than 50% of the initial value.

With the new version of the boxplot (Figure 4) it clearly noted that the low value is due to an outlier observation. We don´t have a clear explanation for the response of that outlier, but it seems that a small variation in the experimental conditions or in the delivery of 13desmSPXC can have contributed to it.

4、Why the concentration of 13-desm SPX in the raw group was lower than that in the boiled group.

The concentration of 13-desm SPX in the raw group was not lower than that in the boiled group.

Round 3

Reviewer 2 Report

As the authors replied, different microbial community exist in different shellfish living in the same region. The ability of different microorganisms in shellfish to transform 13-desm SPX may vary in shellfish, and therefore the concentration of isomer varies in different shellfish. Therefore, the role of microorganisms may be present in the transformation of 13-desm SPX.

Although three data can be computed to plot a box plot, the plot is not scientifically meaningful. Box plots visually show the distribution of numerical data and skewness through displaying the data quartiles (or percentiles) and averages. It is not appropriate to plot a box plot with only three data.

Author Response

Reviewer 2

1.As the authors replied, different microbial community exist in different shellfish living in the same region. The ability of different microorganisms in shellfish to transform 13-desm SPX may vary in shellfish, and therefore the concentration of isomer varies in different shellfish. Therefore, the role of microorganisms may be present in the transformation of 13-desm SPX.

The following paragraph has been added in line 141:

“Even when the microbial origin of the responsible enzymes cannot be discarded, three facts make it unlike. The first one is that the organisms used in this experiment were maintained for 7 days in filtered seawater, which reduces substantially the organic contents of the gut and, hence, the microbial load. The second is that, in a yet unpublished study (submitted), large differences in transformation among different bivalve species sharing the same habitat have been found. The third is that spirolides are accumulated intracellularly which would require intracellular bacteria to carry out the transformations recorded in natural bivalve populations.”

2.Although three data can be computed to plot a box plot, the plot is not scientifically meaningful. Box plots visually show the distribution of numerical data and skewness through displaying the data quartiles (or percentiles) and averages. It is not appropriate to plot a box plot with only three data.

The Box plot in figure 4 has been modified.
